# Prevalence of *Histoplasma* Antigenuria among Outpatient Cohort with Advanced HIV in Kampala, Uganda

**DOI:** 10.3390/jof9070757

**Published:** 2023-07-18

**Authors:** Preethiya Sekar, Elizabeth Nalintya, Richard Kwizera, Claudine Mukashyaka, Godfrey Niyonzima, Loryndah Olive Namakula, Patricia Nerima, Ann Fieberg, Biyue Dai, Jayne Ellis, David R. Boulware, David B. Meya, Nathan C. Bahr, Radha Rajasingham

**Affiliations:** 1Infectious Diseases Institute, Makerere University, Kampala P.O. Box 22418, Uganda; sekar006@umn.edu (P.S.); enalintya@idi.co.ug (E.N.); rkwizera@idi.co.ug (R.K.); cmukashyaka@idi.co.ug (C.M.); gniyonzima@idi.co.ug (G.N.); lnamakula@idi.co.ug (L.O.N.); pnerima@idi.co.ug (P.N.); david.meya@gmail.com (D.B.M.); 2Division of Infectious Diseases and International Medicine, Department of Medicine, University of Minnesota, Minneapolis, MN 55455, USA; afieberg@umn.edu (A.F.); biydai@umn.edu (B.D.); boulw001@umn.edu (D.R.B.); 3Clinical Research Department, London School of Hygiene and Tropical Medicine, London WC1E 7HT, UK; j.ellis@doctors.org.uk; 4Division of Infectious Diseases, University of Kansas, Kansas City, KS 66160, USA; nate.bahr@gmail.com

**Keywords:** histoplasmosis, prevalence, opportunistic infection, advanced HIV disease

## Abstract

In sub-Saharan Africa, an estimated 25% of people with HIV present with advanced HIV and are at high risk of opportunistic infections. Whereas histoplasmosis has occasionally been seen in Uganda, the understanding of the local risk of acute infection is limited. We sought to determine the prevalence of *Histoplasma* antigenuria using an enzyme immunoassay (EIA, clarus *Histoplasma* GM EIA, IMMY; Norman, OK, USA) in a cohort of outpatients with advanced HIV disease in Kampala, Uganda. Among the persons with positive urine *Histoplasma* antigen tests, we assessed their clinical presentation and outcomes. The EIA was run on stored urine samples as per the manufacturer’s instructions. Specimens ≥1 EIA units were considered positive. Among the 388 tested urine samples, 4 (1.2%) were positive for *Histoplasma* antigen. The histoplasmosis prevalence among participants with a CD4 < 100 cells/mcL was 2.5% (4/158). Three of the four participants with a positive *Histoplasma* antigen test reported systemic symptoms consistent with histoplasmosis. All four participants had a positive urine lipoarabinomannan test and were treated for tuberculosis. By the four-week follow-up visit, all participants were clinically improved, alive, and in care without antifungal therapy. In advanced HIV, the clinical presentations of tuberculosis and histoplasmosis overlap. The value of histoplasmosis screening and pre-emptive treatment is an area of future research.

## 1. Introduction

Histoplasmosis is an endemic dimorphic fungal infection caused by *Histoplasma capsulatum*. *H. capsulatum var. capsulatum (Hcc)* is a subtype of histoplasmosis that predominates in North, Central, and South America, as well as in parts of Asia and Africa [1]. *H. capsulatum var. duboisii* (*Hcd*) was first discovered in Africa in 1942 and has only been reported within the continent (primarily west Africa) or in travelers coming from Africa [1]. Amongst the people with HIV (PWH), histoplasmosis most frequently presents as a disseminated infection with associated fever, fatigue, malaise, anorexia, weight loss, and respiratory symptoms. Progressive disseminated histoplasmosis is an AIDS-defining illness that has historically been described as fatal in the absence of appropriate antifungal treatment. Given the nonspecific nature of the clinical presentation and radiological features of HIV-associated histoplasmosis, it is often mistaken for tuberculosis (TB) [2,3,4,5].

Histoplasmosis is a common infection in the Americas, with a reported prevalence of 22–30% in populations with AHD [6,7]. Recent data suggest that the geographic distribution of histoplasmosis is broader than previously thought, including in places that are not often considered endemic, such as Uganda [8]. However, the true global burden of histoplasmosis remains uncertain. In part, this is due to the poor diagnostic capacity for the disease [9,10,11]. Previously, conventional laboratory methods including culture and histopathological or cytology on tissues were the main methods used to diagnose histoplasmosis [11]. The utility of culture is limited by the long turnaround time and biosafety requirements. While histopathology results are timelier than those of culture, histopathology has low sensitivity and requires skilled professionals to perform and interpret the testing; thus, such testing is frequently not possible in many low-resource settings. Other options for serological and urine histoplasmosis testing include complement fixation, immunodiffusion, and enzyme immunoassays. Complement fixation assays detect antibodies that appear 3–6 weeks after infection in 95% of infection cases but can persist for years. Immunodiffusion tests are able to detect both chronic and acute histoplasmosis. The sensitivity of these tests ranges from 70 to 90%, and specificity is 70–100%, with the immunodiffusion tests being slightly more specific than complement fixation assays [12]. Enzyme immune assays, which target antigen detection, are particularly useful in acute disease settings and in those in immunocompromised states given the lack of antibody production. *Histoplasma* antigen testing, with sensitivity ranging from 67 to 91% and specificity ranging from 91 to 98% [7,13], which may improve the detection of early infection and reduce the time to treatment, especially in low-resource settings [14]. Traditionally, antigen testing has only been available in resource-rich settings but access to antigen tests is improving.

The first population-wide studies of histoplasmosis were conducted with histoplasmin skin tests. Within Africa, skin sensitivity positivity ranges from 0 to 35% [15]. The prevalence of histoplasmin-positive skin tests ranges from 0 to 10% across different districts in Uganda, with the highest prevalence being 12% in the Busoga region of eastern Uganda [16]. While these data give us an estimate of *Histoplasma* positivity, the data were collected prior to the beginning of the AIDS epidemic. Recently, a cohort of patients with suspected cryptococcal meningitis in Kampala, Uganda, was found to have no cases of histoplasmosis antigen positivity and 1.3% (2/151) of these were antibody-positive [17]. However, an outpatient cohort presenting for HIV care is likely more generalizable regarding histoplasmosis screening, prevalence, and outcomes in HIV programs.

The objective of this study was to report the prevalence, clinical presentation, and outcomes of outpatients with advanced HIV disease screened for *Histoplasma* antigen in Kampala, Uganda.

## 2. Materials and Methods

### 2.1. Study Design and Setting

This prospective cohort study was nested within the An ENhanced package of Care tO REduce mortality in advanced HIV disease (ENCORE) trial (clinicaltrials.gov NCT05085171). The ENCORE trial is an ongoing cluster randomized trial evaluating an enhanced package of care for screening and pre-emptive therapy for opportunistic infections among persons with advanced HIV disease. Enrollment began in May 2022. The enhanced treatments include isoniazid + rifapentine (1HP) for latent TB treatment and fluconazole + flucytosine for persons with cryptococcal antigenemia and high titers ≥ 1:160. Participating clinics (seven public and one private) are in urban and periurban settings within the Kampala and Wakiso districts of Uganda.

### 2.2. Study Participants and Procedures

ENCORE participants are adults (≥18 years old) with advanced HIV disease (defined as CD4 cell count < 200 cells/μL). From this cohort, baseline urine samples were obtained from consecutive, consenting ENCORE participants irrespective of clinical symptoms and signs. Urine samples were stored at −80 °C. Samples were thawed and batched to run on an enzyme immunoassay (EIA) (Clarus *Histoplasma* GM EIA (Immy, Norman OK), following the manufacturer’s instructions for the calibrator cut-off procedure. Specimens that were above the cutoff (≥1 EIA Units) were determined to be positive *Histoplasma* antigen cases. There was no reference standard used as this was not a diagnostic accuracy study. IRB approval was obtained through the University of Minnesota, the Infectious Diseases Institute, and the Uganda National Council for Science and Technology.

### 2.3. Data Collection

Baseline clinical and laboratory parameters were collected as part of the parent ENCORE trial. At baseline, the data collected in this study were entered and maintained in REDCap (Vanderbilt, Nashville, TN, USA). The data collected included baseline demographics, CD4 cell count, urine TB lipoarabinomannan (LAM) results, and cryptococcal antigen testing results. Clinical symptoms of cough, fever, weight loss, and night sweats were collected at baseline on all participants. Concurrent medications including anti-TB medications and antifungals were recorded for all participants. Participant vital status was documented over six months.

### 2.4. Data Analysis

We estimated the prevalence of histoplasmosis in our population by evaluating the number of participants positive for urine *Histoplasma* antigen via EIA divided by the total number of samples tested. Summary statistics were analyzed using SAS version 9.4. Continuous variables were summarized as medians and interquartile range (IQR). Categorical variables were summarized as number (N) and percentage. Given the low number of *Histoplasma* antigen-positive prevalence in our cohort, significance testing was not performed. Among participants with a positive *Histoplasma* antigen test, medical charts were reviewed for the presence of pulmonary symptoms, systemic symptoms, diagnosis of TB (presumptive and confirmed), and vital status.

## 3. Results

A total of 388 participants’ urine samples were tested for *Histoplasma* antigen, and 4 (1.2%) were positive. The positive sample values ranged from 2.6 to 4 EIA units. The baseline characteristics of the study population are summarized in Table 1. In the *Histoplasma*-negative group, the median age was 33 years (IQR: 28–40), 56% were ART-naïve, and the median CD4 cell count was 122 cells/μL (IQR: 50–193). *Histoplasma*-positive participants had a median age of 43 years (IQR: 31–45), 75% were ART-naïve, and they had a median CD4 cell count of 39 cells/μL (IQR: 16–54). The prevalence among participants with a CD4 < 100 cells/μL was 2.5% (4/158) (Table 2), and no participants with CD4 > 100 cells/μL were positive. All four people in our cohort with *Histoplasma* antigenuria had a positive TB urine LAM test; all received anti-TB therapy while enrolled in the study. Two cases presented symptomatically with a history of cough, weight loss, and physical weakness; one had only weight loss; and the other had no pulmonary or systemic symptoms. None of the participants with *Histoplasma* antigenuria were cryptococcal-antigen-positive.

No antifungal therapy was given to participants with a positive *Histoplasma* antigen test, as the test was run on stored urine. One participant completed a 6-month follow-up. Two participants completed 12 weeks of follow up and were asymptomatic by that time. The fourth participant completed four weeks of follow up and was asymptomatic.

### 3.1. Histoplasma Antigenuria Case Presentations

#### 3.1.1. Case #1

A 40-year-old female presented to the clinic with a 6-month history of weight loss, weakness, intermittent fevers, night sweats, and a productive cough without hemoptysis. She had been diagnosed with HIV six months prior to presentation and was ART-naïve. She had a CD4 cell count of 27 cells/μL. Her *Histoplasma* antigen count was 44.1 EIA units, urine TB LAM was positive (2+), and CRP was 33 mg/L.

On examination, she was afebrile and weighed 52 kg. She was started on first-line TB treatment (rifampin/isoniazid/pyrazinamide/ethambutol) based on her symptoms and positive urine TB LAM. On her day 14 follow-up visit, she reported improvement and her the systemic symptoms had resolved, although she reported ongoing joint pain. At 14 days, she was initiated on ART (tenofovir/lamivudine/dolutegravir) and was advised to continue taking her TB medications. She was last seen by the study team at her 6-month visit, at which time she reported feeling improved, though still reported general body weakness and joint pains.

#### 3.1.2. Case #2

A 45-year-old female was diagnosed with HIV on the day of presentation. Thus, she was ART-naïve. She reported a two-week history of cough without hemoptysis, one month of fevers, and general body weakness, unintentional weight loss, and wasting. She denied chest pain and shortness of breath. She had no prior history of TB. On examination, she was afebrile and weighed 40 kg. Workup revealed a CD4 count of 5 cells/μL, *Histoplasma* antigen was 33.7 EIA units, CRP was 58 mg/L, and 1+ positive urine TB LAM. She was started on first-line anti-TB therapy. Two weeks after the initial visit, her cough and fever had resolved and she was initiated on ART (tenofovir/lamivudine/dolutegravir). The patient is currently enrolled in the study and is clinically stable 12 weeks after enrollment.

#### 3.1.3. Case #3

A 45-year-old male was diagnosed with HIV one month prior to presentation. He was previously on ART but discontinued therapy 3 weeks prior to presentation. He presented with a prior history of confirmed TB via urine LAM and was receiving first-line TB treatment. At presentation, he was afebrile without cough, night sweats, fevers, or shortness of breath. The initial CD4 count was 51 cells/μL; *Histoplasma* antigen was 2.64 EIA Units. He was asymptomatic at his one-week visit and was initiated on ART (tenofovir/lamivudine/dolutegravir). He was most recently seen 12 weeks after enrollment without new symptoms and was clinically stable.

#### 3.1.4. Case #4

A 21-year-old ART-naïve woman presented two days after being diagnosed with HIV infection. She reported weight loss but denied cough, night sweats, or fevers. She had no prior history of TB and had never received any TB preventive therapy. She was afebrile, and her physical exam was notable for wasting. CD4 cell count was 57 cells/μL, *Histoplasma* antigen was 40 EIA units, TB urine was LAM-positive, and CRP was 24 mg/L. She was started on first-line anti-TB medications. At the day 14 visit, she was started on ART (tenofovir/lamivudine/dolutegravir). The participant has completed 4 weeks of follow-up without any new symptoms and a 2 kg increase in weight.

## 4. Discussion

In our cohort of outpatients with advanced HIV disease, among those with PWH with CD4 < 100 cells/μL, 2.5% had *Histoplasma* antigenuria, and among all those with advanced HIV disease, the *Histoplasma* antigen prevalence was 1.2%. Three out of four presented with symptoms that overlapped between histoplasmosis and TB (unintentional weight loss, cough, and fevers). All had a positive urine TB LAM and received anti-TB therapy. Of note, the urine TB LAM test has a 98% specificity and 65% sensitivity among people with HIV. Cross-reactivity with the *Histoplasma* EIA has been reported with other dimorphic fungi, such as Blastomycoses, Sporothrix, Paracoccidiodes, and Blastomyces. However, no studies have demonstrated cross-reactivity between *Histoplasma* EIA and tuberculosis, though cross-reactivity was previously shown between *Histoplasma* serologic testing (immunodiffusion and complement fixation) and tuberculosis [18,19,20,21]. None had received antifungal therapy, and all four participants were alive and in care with resolution of their symptoms. This is in direct contrast to prior descriptions of progressive disseminated histoplasmosis of 100% mortality if untreated [22]. This either reflects effective and rapid immune recovery with integrase-inhibitor-based regimen controlling histoplasmosis or false-positive urine antigen results. However, that the antigen tests solely occurred in those with CD4 < 100 cells/μL coupled with compatible symptomatology makes dismissing the results problematic. Our study also highlights the complexity of diagnosis and management of co-infections in persons with advanced HIV disease.

The prevalence of histoplasmosis has been evaluated in other African settings. Previously, in Uganda, a seroprevalence study of *Histoplasma* antigen found no positive *Histoplasma* antigen results in 157 urine, serum, and CSF samples tested among people with HIV and cryptococcal meningitis [17]. In other sub-Saharan African countries, *Histoplasma* antigenuria incidence was 4.7% in Ghana [23], 1.1% in Tanzania [24], 7.7–12.7% in Nigeria [25,26], 26% in Cameroon [27], and 14% in South Africa [28]. This variable prevalence has driven the World Health Organization (WHO) and Pan-American Health Organization (PAHO) to include histoplasmosis urine antigen testing as well as treatment in their advanced HIV guidelines.

The detection of *Histoplasma* galactomannan antigen using the EIA has been validated for use with serum, urine, and other body fluids, such as cerebrospinal fluid (CSF) and BAL fluid [29]. The sensitivity with urine varies by disease type but is estimated to be between 79% and 98% [30,31,32]. The IMMY EIA urine test was designed for the detection of galactomannan antigen for the variant *H. capsulatum var capsulatum*. In African countries where both fungal variants have been isolated, there is a possibility that those infected with *H. capsulatum var duboisii* may be missed, hence the low prevalence [33]. The cross-reactivity for the two variants needs to be assessed.

Our study was limited by the small number of positive results found, which prevented further statistical analyses to compare the characteristics of patients found to be *Histoplasma* antigen-positive and -negative. We described in case #3 an asymptomatic person who tested positive for *Histoplasma* antigenuria with borderline positive EIA units. In these cases, it would be appropriate to test for other fungal infections to assess for cross-reactivity; such tests were unavailable in our setting. Culture was not performed to confirm histoplasmosis. Additionally, our clinics were located in semiurban areas in the Kampala and Wakiso districts of Kampala. Across Uganda, there is likely variability in *Histoplasma* antigen prevalence [16].

In this paper, we described the four patients who were *Histoplasma* antigen-positive, and all presented with pulmonary or systemic symptoms of histoplasmosis. As the diagnosis of histoplasmosis was made retrospectively, none of the patients received antifungal treatment. All *Histoplasma* antigen-positive participants recovered fully from their respiratory and systemic symptoms possibly because histoplasmosis can be self-limiting in milder disease states and the patients may have cleared their infection with immune reconstitution of ART alone. Alternatively, some results may have been false positives and clinically irrelevant. Given all four patients with positive tests had positive urine LAM tests, one could posit false positivity due to LAM. More studies among persons without TB co-infection may shed light on the presenting symptoms and resolution with and without antifungal therapy. Biomarkers that predict TB versus histoplasmosis may be helpful given the overlapping clinical presentations. For example, in French Guiana, using a multivariate model, gastrointestinal symptoms was more predictive of histoplasmosis compared with TB [34]. Further research to better understand the outcomes of patients who test *Histoplasma* antigen-positive but remain asymptomatic and the role of pre-emptive therapy is warranted. In a setting where the prevalence might be low and patients remain largely asymptomatic, it is important to understand the costs and benefits of health screening programs.

## Figures and Tables

**Table 1 jof-09-00757-t001:** Baseline characteristics by *Histoplasma* antigen status.

	*Histoplasma* Antigen-Negative	*Histoplasma* Antigen-Positive
Total number	384	4
Age, years	33 (28–40)	43 (31–45)
Female	202 (53%)	3 (75%)
Weight, kg	56 (49–61)	45 (40–52)
CD4 cell count, cells/μL	122 (50–193)	39 (16–54)
CRP, mg/L	5 (1–29)	29 (14–46)
ART-naïve	216 (56%)	3 (75%)
ART-experienced	168 (44%)	1 (25%)
Urine TB LAM positive	119 (32%)	4 (100%)
Symptoms of active TB *	123 (32%)	2 (50%)
Receiving active TB treatment	146 (38%)	4 (100%)

Continuous variables are summarized as medians and interquartile range (IQR). Categorical variables are summarized as number (N) and percentage. * Cough, fever, weight loss, or night sweats. Abbreviations: kg = kilogram, μL = microliter, CRP = C reactive protein, mg = milligram, L = liter, ART = antiretroviral therapy, TB = tuberculosis, LAM = lipoarabinomannan.

**Table 2 jof-09-00757-t002:** *Histoplasma* antigen prevalence stratified by CD4 cell count.

CD4 Cell Count (cells/µL)	*Histoplasma* Antigen-Negative	*Histoplasma* Antigen-Positive
<50	92	2
50–100	66	2
101–150	60	0
151–200	68	0
>200	87	0

## Data Availability

Research data are available as a supplemental table to this study.

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
