# Peer review of "Prevalence of *Histoplasma* Antigenuria among Outpatient Cohort with Advanced HIV in Kampala, Uganda"

_jof, 2023, doi:10.3390/jof9070757_

Round 1
Reviewer 1 Report
Please, see the attached document.

Reviewer 2 Report
The article by Sekar et al., is a study of the prevalence of histoplasmosis in advanced HIV patients. Since histoplasmosis is a neglected disease in Africa, that is probably underdiagnosed, this study seems me interesting. The articles is well written and results are clear. Just some comments:
In my opinion trying to confirm histoplasmosis in the positive patients would have been very interesting. Other techniques such as PCR could have been a good option. Didn't the authors consider this option?
Rule out the possibility of that H. capsulatum var duboisii strains do not work in the antigen test is also interesting. An “in vitro” assay could have been interesting.
